# MR T2* Map to Predict Worsening Hypertension Control: A Preliminary Study

**DOI:** 10.3390/life15010073

**Published:** 2025-01-09

**Authors:** Chun-Hung Liu, Jeon-Hor Chen, Antonio Carlos Westphalen, Chun-Ming Chen, Chih-Ping Chang, Wei-Ching Lin

**Affiliations:** 1Department of Medical Imaging, China Medical University Hospital, Taichung 40402, Taiwan; jjaymax@hotmail.com (C.-H.L.); jinmingc@yahoo.com.hk (C.-M.C.); 2Department of Radiology, E-Da Hospital and I-Shou University, Kaohsiung 82445, Taiwan; jeonhc@uci.edu; 3Department of Radiological Sciences, University of California, Irvine, CA 92697, USA; 4Departments of Radiology, Urology and Radiation Oncology, University of Washington, Seattle, WA 98195, USA; acwestph@uw.edu; 5Department of Cardiology, China Medical University Hospital, Taichung 40402, Taiwan; 007466@tool.caaumed.org.tw; 6Department of Medicine, School of Medicine, China Medical University, Taichung 40402, Taiwan; 7Department of Biomedical Imaging and Radiological Science, School of Medicine, China Medical University, Taichung 40402, Taiwan

**Keywords:** T2* mapping, hypertension, blood pressure, ARB, renal oxygenation

## Abstract

Blood pressure measurement is important in monitoring hypertension. However, blood pressure does not provide much information about renal condition in treated hypertension. This study aimed to evaluate renal oxygenation in hypertensive patients using T2* mapping. Subgroup analysis explored whether R2* values can guide adjustments in antihypertensive treatment. A total of 140 consecutive subjects were recruited: 87 hypertensive subjects and 53 normotensive subjects. Hypertensive subjects were classified into non-medication (non-med), angiotensin II receptor blocker (ARB), and non-ARB-treated groups. Each group was divided into good and poor control subgroups based on blood pressure at enrollment. T2* mapping was utilized to assess renal cortical and medullary R2* values. After a 2-year follow-up, subjects were categorized into stable and unstable based on the need for treatment modifications. The unstable subgroup had higher medullary R2* values than the stable subgroup in all followed patients (*p* < 0.05). Additionally, the unstable merged non-med with ARB subgroup had higher medullary R2* values overall (*p* < 0.05) and within the good control subgroup (*p* < 0.05). Patients with stable hypertension, especially those with good control managed through lifestyle modifications or ARBs, exhibited lower renal medullary R2* values, suggesting higher renal oxygenation.

## 1. Introduction

Hypertension can cause renal hypoxia, which is vital in the pathogenesis of chronic kidney disease (CKD) [1]. This can lead to a vicious cycle, as renal hypoxia can induce further increases in blood pressure (BP) [2].

The oxygenation of the kidney is influenced by renal blood flow and oxygen consumption, which in turn depends on sodium reabsorption in renal tubules [2,3]. The interaction between nitric oxide and angiotensin II is crucial in regulating renal perfusion, glomerular filtration, and tubular function. Nitric oxide promotes renal vascular smooth muscle relaxation, maintaining blood flow and glomerular filtration, while angiotensin II increases vascular resistance and sodium reabsorption [4]. In hypertension, endothelial dysfunction reduces nitric oxide availability, leading to increased angiotensin II production, decreased renal blood flow, and impaired oxygenation [2,5,6]. Due to its significant role in hypertension development [7], renin–angiotensin–aldosterone system (RAAS) inhibitors are typically the first-line treatment choice for improving renal oxygenation [8]. Other antihypertensive drugs like beta-blockers, calcium channel blockers (CCB), and diuretics also impact these renal regulatory mechanisms but have varying effects on oxygenation [9,10,11].

Hypertensive patients often undergo a comprehensive follow-up strategy that combines home BP measurements, daily logs, 24 h monitoring, or repeated measurements at the office [12]. Despite well-controlled BP, renal function may worsen due to poor oxygenation [13]. Monitoring renal oxygenation could help prevent the vicious cycle of renal hypoxia-induced increasing BP. However, direct measurement of renal oxygenation using methods like microelectrode-based techniques is invasive and requires inserting electrodes directly into renal tissue, a procedure primarily used in animal studies [14].

T2* mapping provides a non-invasive way to monitor tissue oxygenation, and it is a gradient echo MRI sequence sensitive to paramagnetic materials like deoxyhemoglobin, which decreases T2* relaxation time and consequently elevates the R2* value [15]. It reflects tissue oxygenation and has been used to detect brain oxygenation changes [16]. Recently, it has become popular for noninvasive renal oxygenation evaluation in renovascular disease, diabetic nephropathy, and CKD [17,18,19]. T2* mapping studies on diabetic patients using RAAS inhibitors at the peak effect time point showed unexpected results, differing from animal studies [18]. Although a coincidental finding was that a single-dose intravenous injection of furosemide can increase renal oxygenation, this does not fit the usage in the clinical scenario [20]. More research is needed on renal oxygenation monitoring and hypertension treatment effects.

This study aimed to evaluate renal oxygenation in treated hypertensive patients using T2* mapping. Additionally, subgroup analysis was conducted to determine whether the R2* value can help clinicians decide the need to adjust antihypertensive regimens.

## 2. Materials and Methods

### 2.1. Subjects

The institutional review board approved this prospective combined case–control and cohort study. From February 2020 to October 2021, patients aged ≥ 18 years who were scheduled for a 1.5 T abdominal MRI (Optima MR450w, GE Healthcare, Milwaukee, WI, USA) referring from out-patient departments, excluding those with renal lesions, were consecutively recruited if they had serum creatinine and estimated glomerular filtration rate (eGFR) data within the past month. Informed consent was obtained from all subjects.

They were divided into hypertensive and normotensive groups. The inclusion criteria for the hypertensive group were: (1) a clinical diagnosis of hypertension according to the 2020 International Society of Hypertension Global Hypertension Practice Guidelines, (2) availability of information on antihypertensive medication. Normotensive subjects are eligible if they have no history of hypertension.

Exclusion criteria for both the hypertensive and normotensive groups included the following: (1) stage 4 or 5 CKD, (2) infiltrative renal lesions, renal arterial stenosis, chronic venous congestion, or ureteral obstruction, (3) renal iron deposition, (4) adrenal gland tumors, and (5) poor MR imaging quality or T2* mapping acquired after contrast injection.

The subjects in the hypertensive group were classified into three subgroups based on their treatment: (1) non-medication (non-med) subgroup—subjects treated with lifestyle modifications only; (2) angiotensin II receptor blocker (ARB) subgroup—subjects treated with ARB alone or with other antihypertensive drugs; and (3) non-ARB subgroup—subjects treated with beta-blockers, CCB, or diuretics, but not ARB. Each subgroup was further divided into good control and poor control based on BP readings. Good control was defined as in-office systolic BP < 140 mmHg and diastolic BP < 90 mmHg, while poor control was office systolic BP ≥ 140 mmHg or diastolic BP ≥ 90 mmHg. The BP was measured at least once with the subject seated and relaxed for at least 5 min with the mid-arm at heart level and resting on the table.

### 2.2. MRI Studies

MRI studies were performed after 4 h of fasting except for current medications.

T2* mapping used a multi-echo gradient echo sequence on a 1.5 T MRI system. Images were obtained before administering intravenous contrast agents. Six to seven coronal sections were obtained to cover both kidneys with a thickness of 5 mm and a gap of 3 mm. The scanning parameters were as follows: repetition time = 65 ms, flip angle = 30°, bandwidth = 41.67 kHz, matrix 256 × 256, and field of view = 40 cm. Fourteen echo time values ranged between 2.5 and 53.2 ms, with 3.9 ms inter-echo spacing. T2* mapping required three 15 s breath holds, with the entire sequence taking less than 1 min, including rest periods.

### 2.3. Imaging Interpretation

T2* mapping images were analyzed by reader 1 (CHL, 1-year experience in abdominal MRI), supervised by reader 2 (WCL, 18-year experience), using a commercially available workstation (Advantage Workstation Volume Share 7, GE Healthcare, Milwaukee, WI, USA). R2* values were measured in six 0.2–0.3 cm^2^ regions of interest and manually placed in each kidney’s cortex and medulla near or at the hilar level at the upper, mid, and lower poles, avoiding vessels, artifacts, and focal lesions. The medulla was recognized as the renal parenchyma between the two adjacent interlobular vessels, and the cortex was the area adjacent to the kidney’s border (Figure 1). If the corticomedullary differentiation was poor, the cortex was defined within 2 mm of the renal border [21].

To determine intra- and inter-observer reliability, 30 subjects were selected using stratified random sampling and reanalyzed by both readers after a one-month interval.

### 2.4. Cohort Follow-Up Evaluation

Subjects with antihypertensive treatment records in our hospital and follow-up for 2 years after the MR examination were enrolled in this follow-up study to investigate how renal oxygenation relates to BP control and drug modification. The non-med and ARB subgroups were combined into a “merged non-med with ARB subgroup”, as both approaches can control BP via the RAAS [22,23]. And the subgroups were subdivided into stable (clinically stable and no modification of the antihypertensive treatment) and unstable (uncontrolled hypertension requiring treatment modification) [8,24].

### 2.5. Statistical Analysis

All statistical analyses were performed using IBM SPSS Statistics, version 20 (SPSS Inc., Chicago, IL, USA). The normality of continuous data were assessed using the Kolmogorov–Smirnov test. Continuous variables with a normal distribution are presented as mean ± standard deviation, while those with a non-normal distribution are presented as median with interquartile range. The baseline subject characteristics of the normotensive and hypertensive groups and subgroups were compared using Student’s *t*-test or Mann–Whitney U test, depending on the distribution, for continuous variables, and the Chi-square test for non-continuous variables. Subgroup analysis used one-way analysis of variance and Mann–Whitney U test.

Inter-observer and intra-observer reliability were assessed with the intraclass correlation coefficient (0.0–0.5, poor reliability; 0.51–0.75, moderate reliability; 0.75–0.9, good reliability; 0.90–1.00, excellent reliability) [25]. A *p*-value < 0.05 was considered statistically significant.

## 3. Results

### 3.1. Subject Characteristics

One hundred and fifty consecutive subjects were enrolled, with 56 in the normotensive group and 94 in the hypertensive group. All subjects underwent an MRI exam, but three subjects in the normotensive group and seven subjects in the hypertensive group were excluded due to poor MRI quality caused by motion or susceptibility artifact (N = 8) and or T2* mapping acquired after intravenous contrast administration (N = 2). The flow chart is shown in Figure 2.

Table 1 presents the subjects’ characteristics. The mean age and serum creatinine levels were lower in the normotensive group than in the hypertensive group, and eGFR was higher in the normotensive group.

### 3.2. Antihypertensive Drugs Used in Each Subgroup

There were 38 subjects in the ARB subgroup, including 7 subjects using ARB as monotherapy, 16 using combination therapy of two different antihypertensive drugs (ARB plus beta-blocker, CCB, or diuretic), and 15 using combination therapy of three or more different antihypertensive drugs. In the non-ARB subgroup, there were 22 subjects, with 3 using only beta-blockers, 12 using only CCB, 2 using only diuretics, and 5 using combination therapy of two different antihypertensive drugs without ARB.

There were 24 subjects treated with monotherapy, and they were all stage one hypertension. Among the 36 subjects using combination therapy of two or more different antihypertensive drugs, 10 subjects had diabetes mellitus, 6 subjects had CKD, 7 subjects had both diabetes mellitus and CKD, 1 had heart failure, 5 had stage one hypertension, treated with monotherapy initially and titrated to combined therapy, and 7 had stage two hypertension.

### 3.3. Intra-Observer and Inter-Observer Reliability for R2* Values Measurement

Intra- and inter-observer reliability for mean R2* values were intraclass correlation coefficients of 0.96 and 0.91, respectively.

### 3.4. R2* Values in Each Subgroup

We found no evidence of difference between the mean cortical R2* value of the normotensive and hypertensive groups (12.74 ± 0.94 s^−1^ vs. 12.73 ± 0.80 s^−1^, *p* = 0.93). The mean medullary R2* value was lower in the hypertensive group (16.61 ± 1.56 s^−1^ vs. 17.48 ± 2.10 s^−1^, *p* = 0.006) (Table 2).

No differences in R2* values were observed between the good control and poor control subgroups of any treatment subgroups (Table 2).

### 3.5. R2* Value and Treatment Modification Follow-Up Study

In the good control non-med, ARB, and non-ARB subgroups, 6, 3, and 2 subjects were lost to follow-up, respectively. In the poor control groups, 15, 12, and 1 subjects were lost to follow-up. The merged non-med with ARB good control subgroup had 14 subjects: 8 stable and 6 unstable. The merged non-med with ARB poor control subgroup had 15 subjects: 10 stable and 5 unstable. Overall, 18 subjects were in the stable subgroup, and 11 were in the unstable subgroup of the merged non-med with ARB subgroup. In the good control, non-ARB subgroup, 4 subjects remained stable, but 2 subjects became unstable. In the poor control non-ARB subgroup, 7 subjects were in the stable subgroup, and 6 were in the unstable subgroup. In total, 11 subjects were in the stable subgroup, and 8 were in the unstable group of the non-ARB subgroup. Finally, 29 were in the stable subgroup, and 19 were in the unstable subgroup of all followed subjects. The flow chart is shown in Figure 3.

The subgroups did not differ in age, gender, creatinine, and eGFR (Table 3).

The comparison of mean medullary R2* values between stable and unstable patient subgroups revealed a lower medullary R2* value in the stable group across all patients (16.18 ± 1.53 s^−1^ vs. 17.33 ± 1.64 s^−1^, *p* = 0.02). Further analysis indicated that this difference persisted in the merged non-med with ARB subgroup (16.20 ± 1.23 s^−1^ vs. 17.60 ± 1.41 s^−1^, *p* = 0.007), but not in the non-ARB subgroup (16.15 ± 2.00 s^−1^ vs. 16.95 ± 1.96 s^−1^, *p* = 0.6) (Table 3).

Comparing the good and poor control subgroups within the merged non-med with ARB subgroup, the mean medullary R2* value was lower in the stable subgroup (16.84 ± 1.00 s^−1^ vs. 17.95 ± 0.75 s^−1^, *p* = 0.03) of the good control subgroup. However, this difference did not reach statistical significance for the poor control subgroup (stable: 15.69 ± 1.20 s^−1^; unstable: 17.19 ± 1.96 s^−1^; *p* = 0.09) (Table 4).

The median time from the MRI exam to the modification of antihypertensive drugs in the unstable group was 15 [8,9,10,11,12,13,14,15,16,17,18,19] months. No patient required a drug modification immediately.

## 4. Discussion

The results of this study show that renal medullary R2* values were lower, and therefore oxygenation presumably higher, in patients with stable hypertension, particularly those with good control when managed with lifestyle modifications or ARBs.

Pruijm et al. found no significant oxygenation change before and after ARB treatment [20]. Their analysis lacked subgroup differentiation by BP control levels, possibly masking variations. This study performed a subgroup analysis based on office BP control and treatment adjustments during the two-year follow-up. The results support the theory that stable hypertension control, primarily through RAAS inhibition, preserves renal medullary oxygenation and negates the need for treatment changes [24].

Approximately 43% of patients who were initially well-controlled with non-medication or ARBs developed unstable hypertension, necessitating treatment changes. Possible reasons include:Masked Uncontrolled Hypertension: Normotensive in-office but high BP outside. Office BP measurements provide a snapshot and do not capture daily variability [24].Disease Progression: Some patients experienced worsening renal function despite controlled BP, suggesting antihypertensive therapies did not reverse renal hypoxia, possibly due to small renal vessel remodeling [13,26,27,28]. Patients with masked uncontrolled hypertension had higher renal function deterioration rates, indicating underlying renal hypoxia [29,30].

Conversely, about 67% of poorly controlled patients in the non-medication or ARB subgroup did not need treatment adjustment, likely due to the white-coat effect [24]. Their treatment remained unchanged after home BP monitoring and laboratory assessments.

Despite no statistically significant difference, the well-controlled group showed mild renal medullary hypoxia compared to the poorly controlled group in the non-medication and ARB subgroups. This possibly due to including patients with masked uncontrolled hypertension and those with good control BP but inadequate reversal of renal hypoxia in the well-controlled subgroup, as well as patients with white-coat hypertension in the poorly controlled subgroup. Monitoring BP control along with both office and home BP is essential. However, certain situations may cause confusion, such as variability in office or home blood pressure, possible white-coat hypertension or masked hypertension, and resistant hypertension that does not respond to conventional drug therapy. Given these conditions, 24 h ambulatory blood pressure monitoring is an excellent tool [31].

The mean renal medullary oxygenation was higher in the stable non-ARB subgroup, but this was not statistically significant. This group used beta-blockers, CCBs, diuretics, and their combinations, which have varying effects on renal oxygenation. Beta-blockers, for instance, affect renin secretion and vasodilation, influencing renal plasma flow differently depending on the drug and its use [9]. CCBs impact renal perfusion and tubular sodium reabsorption [10], while diuretics affect electrolyte reabsorption in the kidney, potentially increasing renal oxygenation. Due to the diversity of non-ARB drugs and their administration methods, their overall impact on renal oxygen levels varies. In addition, the length of drug use could affect the observed effects, including ARB and non-ARBs. Further research is needed to clarify these effects.

MRI, specifically the T2* map, could help detect masked uncontrolled hypertension and predict worsening hypertension, prompting early treatment changes. However, due to the high cost, it is impractical to offer MRI to everyone. This study did not identify predictors for selecting patients who would benefit from a T2* map, partly due to the small cohort size and lack of correlation analysis with various factors. Further research with a larger sample size, particularly with patients with high BP variability [32], is needed to determine predictive factors and validate the use of T2* maps.

The results showed higher renal medullary oxygenation in the hypertensive group compared to the normotensive group. Subjects restricted water and food intake before the MRI exam, leading to relative dehydration, which can reduce renal blood flow and activate RAAS [33], increasing renal oxygen consumption. Antihypertensive medications, which were not stopped before the MRI, may have influenced renal oxygenation [20,34,35]. In patients treated with ARBs who experience dehydration due to fasting before MRI, dehydration activates the RAAS, which is inhibited by ARBs [33]. Therefore, the higher renal medullary oxygenation observed in the hypertensive group may be attributed to the effect of antihypertensive drugs on RAAS, which reduces dehydration-induced oxygen consumption in the kidneys.

Pruijm et al. found no difference in renal oxygenation between hypertensive and normotensive subjects [36]. They did not restrict concomitant medication and ensured subjects were well-hydrated before the MRI exam. Their normotensive subjects were younger than the hypertensive patients, similar to this study. Prasad et al. indicated that optimal hydration could enhance medullary oxygenation among young participants [37]. Thus, the comparable oxygenation levels in both groups observed in Pruijm et al.’s study suggest the possibility of renal medullary hypoxia in the normotensive group compared to the hypertensive group in this study due to the lack of hydration.

Subjects in the hypertensive group were older and exhibited lower eGFR compared to the normotensive group, which is consistent with the trend of hypertension being more prevalent in the elderly and contributing to CKD, as evidenced by reduced eGFR. This factor could also impact renal oxygenation status [19,36]. Further studies with age-matched subgroup analysis are needed.

This study has some limitations. Firstly, it was conducted at a single center using a single 1.5T MRI, which may affect the generalizability of our results. Further investigations are needed to validate our findings, focusing on R2* values and imaging standardization across multiple sites and scanners. Secondly, placing regions of interest in the kidney for patients with a GFR < 60 mL/min/1.73 m^2^ is challenging due to poor corticomedullary differentiation. However, inter-observer and intra-observer reliability were high when observers followed predefined criteria for identifying the renal medulla and cortex. Recently, AI techniques, such as deep convolutional neural networks with or without transformative-based noise reduction, can be applied to the T2* map to improve image quality. Future research on this new AI application is needed to determine how it improves image quality and if it can make T2* value measurement more effortless and reproducible. Thirdly, the small sample size in subgroups limited the reliability and precluded some analyses, such as multivariate analysis, to identify confounding factors such as age, gender, creatinine/eGFR, underlying diseases, and multi-drug interactions to isolate the proper relationship between renal oxygenation and blood pressure control. Finally, the follow-up period was only two years. It may be too little time to fully understand the relationship between T2* values and long-term blood pressure control outcomes. However, extending it could introduce additional variables, thus complicating interpretation.

## 5. Conclusions

Patients with stable hypertension, especially those with good control managed with lifestyle modifications or ARBs, exhibited lower medullary R2* values, suggesting potentially higher oxygenation. Further research is needed to validate the utility of T2* mapping for assessing renal oxygenation in hypertensive patients and its impact on adjusting hypertension treatment in clinical settings, particularly in managing cases with large variability in office or home blood pressure measurements, as well as conditions like white-coat hypertension, masked hypertension, and resistant hypertension.

## Figures and Tables

**Figure 1 life-15-00073-f001:**
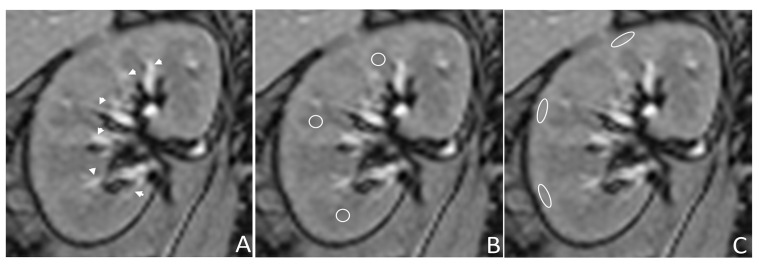
T2* mapping of the right kidney of a 48-year-old male in the normotensive group. (**A**) The interlobar vessels (arrowheads) are seen as linear hyperintense structures in the renal sinus, abutting parenchyma, with radial direction from the hilum. (**B**) The renal medulla’s ROIs (round circles) were placed between the interlobar vessels in the upper, mid, and lower poles. (**C**) The ROIs (oval circles) of the cortex were placed adjacent to the border of the kidney. ROI: region of interest.

**Figure 2 life-15-00073-f002:**
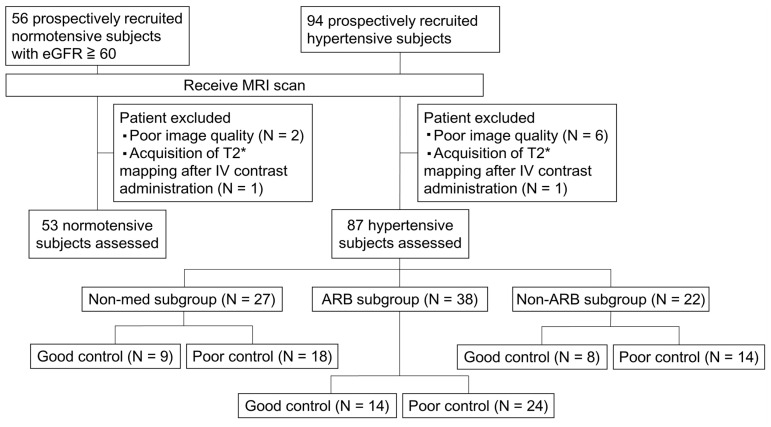
Study flowchart. eGFR, estimated glomerular filtration rate; IV, intravenous; ARB, angiotensin II receptor blocker.

**Figure 3 life-15-00073-f003:**
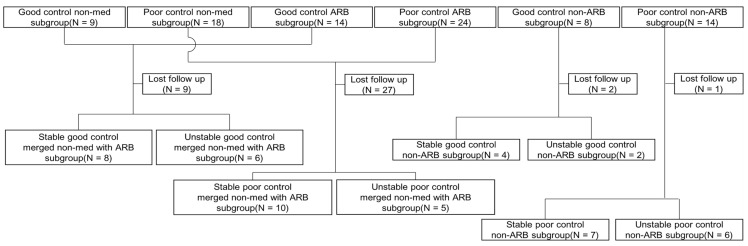
Flowchart of follow-up study. ARB, angiotensin II receptor blocker.

**Table 1 life-15-00073-t001:** Basic information of the normotensive group and hypertensive group.

	Normotensive Group (*n* = 53)	Hypertensive Group (*n* = 87)	*p*-Value
Age (years)	52.5 ± 13.1	64.9 ± 10.3	<0.001 *
Gender ratio (Male/Female)	29/24	59/28	0.12
Creatinine (mg/dL)	0.77 [0.70–0.93]	0.94 [0.79–1.18]	<0.001 *
eGFR (mL/min/1.73 m^2^)	90.1 ± 17.0	74.8 ± 20.9	<0.001 *

eGFR, estimated glomerular filtration rate; * *p* < 0.05.

**Table 2 life-15-00073-t002:** R2* values in the normotensive and hypertensive groups and each hypertensive treatment subgroup.

	Normotensive Group	Hypertensive Group	*p*-Value	Hypertensive Subgroups						
				Non-Med Subgroup		ARB Subgroup		Non-ARB Subgroup		*p*-Value
				Good Control	Poor Control	Good Control	Poor Control	Good Control	Poor Control	
CR2*	12.74 ± 0.94	12.73 ± 0.80	0.93	13.07 ± 0.95	12.82 ± 0.46	12.75 ± 0.80	12.37 ± 0.80	12.62 ± 0.71	13.04 ± 0.94	0.102
MR2*	17.48 ± 2.10	16.61 ± 1.56	0.006 *	17.46 ± 1.28	16.41 ± 1.47	17.12 ± 1.03	16.12 ± 1.36	16.41 ± 1.26	16.76 ± 2.40	0.215

ARB, angiotensin II receptor blocker; CR2*, cortical R2* value; MR2*, medullary R2* value; * *p* < 0.05.

**Table 3 life-15-00073-t003:** Basic information and medullary R2* value in all follow-up hypertensive subgroups.

	Follow-Up Hypertensive Subgroups								
	All			Merged Non-Med with ARB Subgroup			Non-ARB Subgroup		
	Stable Subgroup (*n* = 29)	Unstable Subgroup (*n* = 19)	*p*-Value	Stable Subgroup (*n* = 18)	Unstable Subgroup (*n* = 11)	*p*-Value	Stable Subgroup (*n* = 11)	Unstable Subgroup (*n* = 8)	*p*-Value
Age (years)	65.4 ± 12.2	67.2 ± 10.6	0.64	63.2 ± 12.1	65.7 ± 9.1	0.71	69.0 ± 12.0	69.3 ± 12.8	0.97
Gender ratio (Male/Female)	21/8	11/8	0.30	13/5	6/5	0.33	8/3	5/3	0.64
Creatinine (mg/dL)	10.5 ± 0.33	0.93 ± 0.34	0.13	1.04 ± 0.34	0.96 ± 0.42	0.28	1.07 ± 0.33	0.89 ± 0.19	0.27
eGFR (mL/min/1.73 m^2^)	72.2 ± 22.0	79.0 ± 21.6	0.3	73.2 ± 21.1	78.1 ± 23.1	0.47	70.5 ± 24.3	80.3 ± 20.7	0.4
MR2*	16.18 ± 1.53	17.33 ± 1.64	0.02 *	16.20 ± 1.23	17.60 ± 1.41	0.007 *	16.15 ± 2.00	16.95 ± 1.96	0.6

ARB, angiotensin II receptor blocker; eGFR, estimated glomerular filtration rate; MR2*, medullary R2* value; * *p* < 0.05.

**Table 4 life-15-00073-t004:** Subgroup analysis of the medullary R2*value in the follow-up merged non-med with ARB subgroups.

	Merged Non-Med with ARB Subgroup					
	Good Control (*n* = 14)			Poor Control (*n* = 15)		
	Stable Subgroup (*n* = 8)	Unstable Subgroup (*n* = 6)	*p*-Value	Stable Subgroup (*n* = 10)	Unstable Subgroup (*n* = 5)	*p*-Value
MR2*	16.84 ± 1.00	17.95 ± 0.75	0.03 *	15.69 ± 1.20	17.19 ± 1.96	0.09

ARB, angiotensin II receptor blocker; MR2*, medullary R2* value; * *p* < 0.05.

## Data Availability

The data included in this study are available upon reasonable request by contact with the corresponding author.

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
