# Peer review of "MR T2* Map to Predict Worsening Hypertension Control: A Preliminary Study"

_life, 2025, doi:10.3390/life15010073_

Round 1

Reviewer 1 Report

Comments and Suggestions for Authors

The authors of the study assessed renal function in terms of blood pressure control. The assessment was conducted using magnetic resonance imaging. I have a few comments, primarily regarding the patient groups themselves. The patients were divided into three groups. A group without hypertension, a group with hypertension, which was further divided into a group treated with sartans, i.e. angiotensin II type I receptor antagonists, and for patients not treated with sartans. And here a few questions immediately arise. Monotherapy is recommended only and exclusively for patients, and this has already been recommended in the guidelines since 2018, for either young patients with stage I hypertension or elderly patients with fragility syndrome. This is the first comment, and the authors would have to address it. A patient suffering from type II diabetes or chronic kidney disease should be treated as part of combined therapy, in accordance with the current guidelines. The lack of this information significantly complicates the assessment of whether the choice of monotherapy was appropriate and in line with the applicable standards. Only information is provided on how many patients received dual or triple therapy.

In addition, the study does not specify in detail whether the patients were in a phase of stable hypertension or in a phase of exacerbations, which could affect the interpretation of the results. It would also be important to take into account any differences in the duration of sartan therapy, because the length of drug use could affect the observed effects. There is no information on comorbidities. Therefore, we do not know whether this group included patients with, for example, heart failure and, if so, which one? With preserved systolic function or with reduced function? In either case, both of these groups of patients may receive flozin, which leads to constriction of the afferent arteriole, which may affect kidney function and changes observed in the magnetic resonance image. Similarly, in the case of other drugs used to treat hypertension in patients with heart failure, I mean here mineralocorticoid receptor antagonists or other drugs that would concern the treatment of these diseases such as diabetes or heart failure

In summary, the work raises an interesting topic, but requires supplementation with key data on the characteristics of patients and the methods of treatment used.

Author Response

Comment 1: The authors of the study assessed renal function in terms of blood pressure control. The assessment was conducted using magnetic resonance imaging. I have a few comments, primarily regarding the patient groups themselves. The patients were divided into three groups. A group without hypertension, a group with hypertension, which was further divided into a group treated with sartans, i.e. angiotensin II type I receptor antagonists, and for patients not treated with sartans. And here a few questions immediately arise. Monotherapy is recommended only and exclusively for patients, and this has already been recommended in the guidelines since 2018, for either young patients with stage I hypertension or elderly patients with fragility syndrome. This is the first comment, and the authors would have to address it. A patient suffering from type II diabetes or chronic kidney disease should be treated as part of combined therapy, in accordance with the current guidelines. The lack of this information significantly complicates the assessment of whether the choice of monotherapy was appropriate and in line with the applicable standards. Only information is provided on how many patients received dual or triple therapy.

Response 1: Thank you for the valuable comment. There were 24 subjects treated with monotherapy, and they were all stage 1 hypertension. Among the 36 subjects using combination therapy of two or more different antihypertensive drugs, 10 subjects had diabetes mellitus, 6 had CKD, 7 had both diabetes mellitus and CKD, 1 had heart failure, 5 had stage 1 hypertension, treated with monotherapy initially and titrated to combined therapy, and 7 had stage 2 hypertension. We added this in the result. [Page 4, Paragraph 4, Line 160-165]

Comment 2: In addition, the study does not specify in detail whether the patients were in a phase of stable hypertension or in a phase of exacerbations, which could affect the interpretation of the results. It would also be important to take into account any differences in the duration of sartan therapy, because the length of drug use could affect the observed effects. There is no information on comorbidities. Therefore, we do not know whether this group included patients with, for example, heart failure and, if so, which one? With preserved systolic function or with reduced function? In either case, both of these groups of patients may receive flozin, which leads to constriction of the afferent arteriole, which may affect kidney function and changes observed in the magnetic resonance image. Similarly, in the case of other drugs used to treat hypertension in patients with heart failure, I mean here mineralocorticoid receptor antagonists or other drugs that would concern the treatment of these diseases such as diabetes or heart failure

Response 2: Thank you for your thoughtful comment. Patients in a phase of stable hypertension or in a phase of exacerbation did affect the interpretation of the renal oxygenation results. We recruited subjects from out-patient departments. Acutely ill patients were not included in this study, and no patient required immediate change medication for BP control. We added this to the material and method.[Page 2, Paragraph 5, Line 74] [Page 4, Paragraph 10, Line 202-203]

The length of drug use could affect the observed effects. It needs longitudinal research targeting different phases of drug use in patients with long-term use of the study medication. [Page 8, Paragraph 6, Line 274-275]

The comorbidities and the medications used for these comorbidities would affect the study results. In this study, 10 subjects had diabetes mellitus, 6 had CKD, 7 had both diabetes mellitus and CKD, 1 had heart failure, 5 had stage 1 hypertension, treated with monotherapy initially and titrated to combined therapy, and 7 had stage 2 hypertension. They did receive other medication for their comorbidities. However, this study had a relatively small sample size, which limited multivariate analysis to clarify the effects of these confounding factors. We added this in the result and limitation. [Page 4, Paragraph 4, Line 160-165] [Page 9, Paragraph 4, Line 318-321]

Reviewer 2 Report

Comments and Suggestions for Authors

Dear authors,

The manuscript is interesting and, in general, fairly well-written.

I have some suggestions to further improve the quality of the manuscript.

I would like to suggest that the authors address these limitations in the article, either by discussing them in the limitations section or, where feasible, by making the appropriate revisions:

1. The study's statistical power is limited by its relatively small sample size, particularly in the subgroup analyses. With only 14-15 subjects in some of the compared groups, the ability to detect meaningful differences is constrained. This raises questions about the reliability and generalizability of some of the findings. The patient classification system based solely on office blood pressure measurements is problematic. Using only in-office readings to categorize patients as having "good" or "poor" control fails to account for phenomena like masked hypertension or white coat hypertension. A more robust approach would have incorporated 24-hour ambulatory blood pressure monitoring.

2. The study lacks a comprehensive multivariate analysis that could help identify confounding factors and better isolate the true relationship between renal oxygenation and blood pressure control. The two-year follow-up period, while substantial, may not be long enough to fully characterize the relationship between T2* values and long-term blood pressure control outcomes. Also, I think authors can discuss future research direction as noise-reducted image usage, by breifly mentioning research like Transformative Noise Reduction: Leveraging Transformer-based Deep Network for Medical Image Denoising. This could help understand noise reduction techniques relevant to T2* mapping image quality.

3. The researchers did not standardize the hydration status of participants before MRI scanning. As they acknowledge, the difference in renal oxygenation between hypertensive and normotensive groups may have been influenced by varying hydration levels rather than disease status or treatment effects. The normotensive and hypertensive groups were not age-matched, with the hypertensive group being significantly older (64.9 vs 52.5 years). This age disparity could confound the results since age impacts renal function independently of hypertension.

4. The study was conducted at a single center using one 1.5T MRI scanner. This limits the generalizability of the findings and provides no data on inter-scanner variability of T2* measurements.

The researchers note difficulties in accurately placing regions of interest for measurements in patients with poor corticomedullary differentiation. While they report good inter-observer reliability, this technical challenge could impact measurement accuracy.

5. Finally, the researchers make assumptions about tissue oxygenation based on T2* values without direct validation through other methods. While T2* mapping is a reasonable proxy for tissue oxygenation, direct confirmation would strengthen the findings.

Thank you for your valuable contributions to our field of research. I look forward to receiving the revised manuscript.

Author Response

Comment 1: The study's statistical power is limited by its relatively small sample size, particularly in the subgroup analyses. With only 14-15 subjects in some of the compared groups, the ability to detect meaningful differences is constrained. This raises questions about the reliability and generalizability of some of the findings. The patient classification system based solely on office blood pressure measurements is problematic. Using only in-office readings to categorize patients as having "good" or "poor" control fails to account for phenomena like masked hypertension or white coat hypertension. A more robust approach would have incorporated 24-hour ambulatory blood pressure monitoring.

Response 1: We appreciate the reviewer’s insightful comment. Using office BP alone to monitor blood pressure control is problematic because it may associated with masked hypertension or white-coat HTN. Therefore, in the follow-up study, we added the results of whether the medication was changed, as medication changes can be referenced along with both office and home BP.

24-hour ambulatory blood pressure monitoring (ABPM) is an excellent tool. It provides a more comprehensive view of blood pressure throughout the day and night, capturing variations that might be missed with home monitoring, although patients may experience discomfort. Its measurements are often taken during non-resting periods, such as during physical activity, which can lead to variability in readings and may not accurately reflect a patient's typical resting blood pressure. In this study, ABPM was not used. Future research could combine the data from ABPM to determine BP control condition and correlate it to renal oxygenation. We added this description to the discussion.[Page 8, Paragraph 5, Line 262-266]

Comment 2: The study lacks a comprehensive multivariate analysis that could help identify confounding factors and better isolate the true relationship between renal oxygenation and blood pressure control. The two-year follow-up period, while substantial, may not be long enough to fully characterize the relationship between T2* values and long-term blood pressure control outcomes. Also, I think authors can discuss future research direction as noise-reducted image usage, by breifly mentioning research like Transformative Noise Reduction: Leveraging Transformer-based Deep Network for Medical Image Denoising. This could help understand noise reduction techniques relevant to T2* mapping image quality.

Response 2:

A comprehensive multivariate analysis is needed to identify confounding factors such as age, gender, creatinine/eGFR, underlying diseases, and multi-drug interactions to isolate the true relationship between renal oxygenation and blood pressure control. However, after grouping potential cross-interactive factors, most outcomes for each predictor variable do not reach the number of 10, a common guideline in statistics to ensure reliable estimation of the model coefficients and minimize the risk of over fitting. This preliminary study is limited by a small sample size, which only allows us to observe a possible relationship between renal oxygenation and blood pressure control. Still, it cannot clearly clarify the true relationship. We address this limitation more clearly on [Page 9, Paragraph 4, Line 318-321]

We appreciate the reviewer’s comment regarding the two-year follow-up period. We agree that blood pressure medications may have long-term effects beyond two years. However, extending the follow-up period beyond two years could introduce additional variables, such as changes in medication regimens, comorbid conditions, and other lifestyle factors, which might complicate the interpretation of the relationship between T2* values and blood pressure control outcomes. We added this to the limitation. [Page 9, Paragraph 4, Line 321-324].

Thank you for informing the concept of applying this new AI technique to this study. T2* map image in our study didn’t use the AI technique, but the commercially available AIRTM Recon DL (GE Healthcare, Waukesha, WI) recently extended its use on T2*map. It includes a deep convolutional neural network to aid in raw data reconstruction. Transformative noise reduction is good for effectively capturing and removing globally distributed noise. It can also gather more comprehensive structural information and enhance the integration of complementary key information in the context of image denoising. It is expected to have better performance on image denoising. We had added this to the discussion. [Page 9, Paragraph 4, Line 314-318].

Comment 3: The researchers did not standardize the hydration status of participants before MRI scanning. As they acknowledge, the difference in renal oxygenation between hypertensive and normotensive groups may have been influenced by varying hydration levels rather than disease status or treatment effects. The normotensive and hypertensive groups were not age-matched, with the hypertensive group being significantly older (64.9 vs 52.5 years). This age disparity could confound the results since age impacts renal function independently of hypertension.

Response 3: Thank you for reinforcing that hydration status would affect renal oxygenation. In the study, all participants undergoing MRI were required to fast for 4 hours, including no water intake, which we mentioned in the method. [Page 3, Paragraph 1, Line 98]

Due to the small sample size, the age-matched study is challenging in this study. As we mentioned in the discussion, it should be performed in future studies. [Page 9, Paragraph 3, Line 305-306]

Comment 4: The study was conducted at a single center using one 1.5T MRI scanner. This limits the generalizability of the findings and provides no data on inter-scanner variability of T2* measurements.

Response 4: Thank you for your valuable comment. We acknowledge that conducting the study at a single center using only one 1.5T MRI scanner limits the ability to generalize the findings across different MRI machines, and we agree that investigating inter-scanner variability is essential for a more comprehensive understanding of the T2* measurements. Several factors, like magnetic field homogeneity or gradient system differences, can contribute to inter-scanner variability in T2* measurements even when using the same field strength scanners. This article is a preliminary study, and future studies involving multi-center trials and data from different MRI scanners will be needed to understand and better quantify the extent of variability in T2* measurements. We address this more clearly in the discussion. [Page 9, Paragraph 4, Line 308-310]

Comment 5: The researchers note difficulties in accurately placing regions of interest for measurements in patients with poor corticomedullary differentiation. While they report good inter-observer reliability, this technical challenge could impact measurement accuracy.

Response 5: Thank you for your thoughtful comment. We have standardized the placement of the ROI. The medulla was identified as the renal parenchyma between the two adjacent inter-lobular vessels, and the cortex was the area adjacent to the kidney’s border. This definition improved inter-observer reliability. This is demonstrated in Figure 1.

Comment 6: Finally, the researchers make assumptions about tissue oxygenation based on T2* values without direct validation through other methods. While T2* mapping is a reasonable proxy for tissue oxygenation, direct confirmation would strengthen the findings.

Response 6: Direct measurement of renal oxygenation through methods like microelectrode-based techniques involves significant challenges. Microelectrode measurements require invasive procedures, such as inserting electrodes directly into renal tissue, primarily used in animal studies. Such techniques are not widely applicable to human subjects due to concerns about patient acceptability and the procedure's invasiveness. Moreover, these methods are technically demanding and difficult to implement in a clinical setting, particularly for long-term monitoring. Therefore, T2* mapping remains a practical and non-invasive alternative in clinical studies, offering a valuable tool for assessing renal oxygenation while minimizing patient risk and discomfort. We add this description in the introduction. [Page 2, Paragraph 2, Line 52-55]

Reviewer 3 Report

Comments and Suggestions for Authors

This study evaluated renal oxygenation in hypertensive patients using T2* mapping. The article is well-structured, the objectives are clear and the results corroborated with figures and discussion.

Minor comments

1.      According to the authors, how feasible is T2* mapping for routine clinical use, especially concerning the availability of MRI facilities and associated costs?

2.      Could T2* mapping be extended to assess oxygenation in other target organs affected by hypertension, such as the heart or brain? If so, two additional lines about this can be added to the discussion section.

3.      Line 278, ‘For example, ARBs inhibit RAAS’, briefly discusses it here.

4.      Provide a suitable reference for lines 278-281

Typo errors suggestions

1.      What is the meaning of line 197? There is no 3.1 section in the article.

2.      In line 217, place a comma or hyphen after ARB.

3.      Line 95, mention the table number.

Author Response

Comment 1: According to the authors, how feasible is T2* mapping for routine clinical use, especially concerning the availability of MRI facilities and associated costs?

Response 1: Thank you for the reviewer’s insightful question. If it is confirmed that T2* mapping can effectively predict blood pressure changes, we believe it could become a valuable tool in clinical settings, particularly in cases of considerable variability in office or home blood pressure measurements, possible white coat hypertension, masked hypertension, and resistant hypertension that does not respond to conventional drug therapy. We add this description in the conclusion.[Page 9, Paragraph 5, Line 330-332]

The T2* mapping procedure takes less than a minute, and its cost reduction should be proportional to a full MRI scan. Additionally, this sequence can be added as an extra examination for patients already undergoing an MRI.

Comment 2: Could T2* mapping be extended to assess oxygenation in other target organs affected by hypertension, such as the heart or brain? If so, two additional lines about this can be added to the discussion section.

Response 2: Thank you for the reviewer’s thoughtful suggestion. T2* mapping is primarily used in the heart to measure iron deposition, particularly in conditions like cardiac hemochromatosis. However, there have been studies incorporating stress tests alongside T2* mapping to detect myocardial infarction, as well as myocardial edema and hemorrhage post-MI. Despite these advances, T2* mapping for heart oxygenation assessments is not a routine clinical practice yet. As for the brain, T2* mapping has been explored in research to detect ischemic core areas, particularly in the context of stroke. However, the application of T2* mapping to detect hypertension change in the brain was not as popular as other sequences. 

Comment 3: Line 278(282), ‘For example, ARBs inhibit RAAS’, briefly discusses it here.

Response 3: Thank you for the reviewer's comment. We have revised this section to make it easier to understand.[Page 9, Paragraph 1, Line 289-293]

Comment 4: Provide a suitable reference for lines 278-281

Response 4: Thank you for the reviewer's comment. We have added a reference number here. [Page 9, Paragraph 1, Line 290]

Comment 5: What is the meaning of line 197? There is no 3.1 section in the article.

Response 5: Thank you for pointing this out. This sentence is from the Life template, which needs to be deleted, and we already corrected it. [page 5]

Comment 6: In line 217, place a comma or hyphen after ARB.  

Response 6: Thank you for pointing this out. We add a comma. [page 5, Line 224]

Comment 7: Line 95, mention the table number.

Response 7: Thank you for the thoughtful reminder to check possible missing. In this paragraph, the word 'table' refers to the surface for placing the arm during blood pressure measurement, not the tabular format.

Round 2

Reviewer 1 Report

Comments and Suggestions for Authors

the authors made corrections and I now consider the manuscript to be suitable for publication.

Reviewer 2 Report

Comments and Suggestions for Authors

All comments have been thoroughly addressed. I extend my gratitude to both the authors and editors for taking my opinions into consideration during the review of this manuscript.